# Neural Message Passing for Multi-Relational Ordered and Recursive Hypergraphs

**Naganand Yadati**
naganand@iisc.ac.in
Department of Computer Science and Automation
Indian Institute of Science
Bangalore, Karnataka, 560012

## Abstract

Message passing neural network (MPNN) has recently emerged as a successful framework by achieving state-of-the-art performances on many graph-based learning tasks. MPNN has also recently been extended to multi-relational graphs (each edge is labelled), and hypergraphs (each edge can connect any number of vertices). However, in real-world datasets involving text and knowledge, relationships are much more complex in which hyperedges can be multi-relational, recursive, and ordered. Such structures present several unique challenges because it is not clear how to adapt MPNN to *variable*-sized hyperedges in them. In this work, we first unify exisiting MPNNs on different structures into G-MPNN (Generalised-MPNN) framework. Motivated by real-world datasets, we then propose a novel extension of the framework, MPNN-R (MPNN-Recursive) to handle recursively-structured data. Experimental results demonstrate the effectiveness of proposed instances of G-MPNN and MPNN-R. The code is available. [1]

## 1 Introduction

Message passing neural network (MPNN) has recently emerged as a successful framework by achieving state-of-the-art performances on many graph-based learning tasks [25]. The message-passing operation in the MPNN framework can be viewed as recursive neighbourhood aggregation, where local neighbourhood messages are aggregated and passed on to neighbouring vertices. MPNN has also recently been extended to multi-relational graphs in which each edge is labelled (and possibly directed), and separately to hypergraphs in which each edge can connect any number of vertices.

However, in real-world datasets involving text and knowledge, relationships are much more complex in which hyperedges can act as vertices recursively in other hyperedges. Hyperedges can also be multi-relational with vertices appearing in a fixed order. We illustrate such structures with a toy example in Figure 1. Multi-relational ordered hypergraphs have been shown to provide more flexible organisation of multi-ary relational facts than multi-relational directed edges and have been a recent research topic of interest [74, 19]. Recursive hypergraphs [55] have been shown to flexibly represent a few sentence types such as claims about claims in natural language (e.g. A claimed that B claimed C). Recursivesly structured hypergraphs are also seen in academic network datasets. Such structures present several unique challenges because it is not clear how to adapt MPNN to *variable*-sized hyperedges in them. Our contributions can be summarised as follows.

- We provide a unified MPNN-style framework, which we call G-MPNN (Generalised-MPNN), for multi-relational ordered hypergraphs. Several notable examples of models can be described using the unified framework

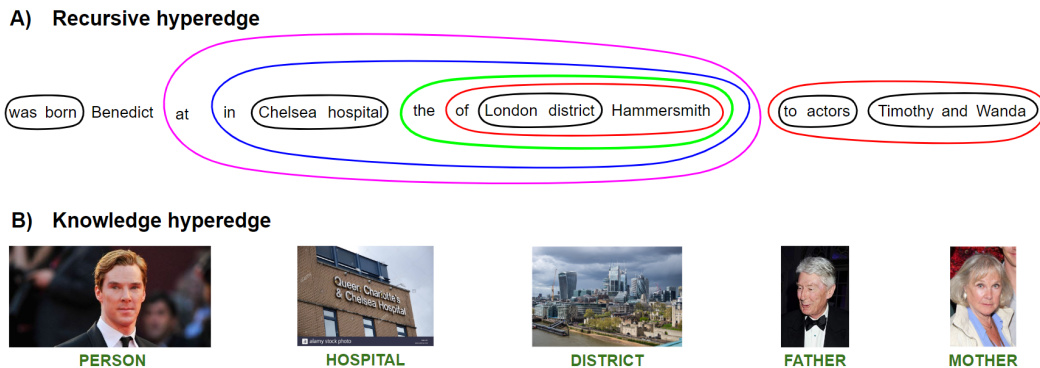

Figure 1: (Best seen in colour) Organisation of text (A) and knowledge (B) for the example sentence: *Benedict was born at Chelsea hospital in the London district of Hammersmith to actors Timothy and Wanda*. **A)** A recursive hyperedge in which hyperedges (shown in different colours) act as vertices in other hyperedges (e.g. black-coloured hyperedge containing *London*, *district* as vertices acts as a vertex in the red-coloured hyperedge containing *of*, *Hammersmith* as vertices). **B)** A 5-ary hyperedge of the relation type: *PERSON was born at HOSPITAL in DISTIRCT to FATHER and MOTHER*.

- Motivated by real-world datasets, we explore the unexplored problem of inductive vertex embedding (embedding unseen vertices at test time) in multi-relational ordered hypergraphs. We deomonstrate the strong inductive capability of G-MPNN on real-world multi-relational ordered hypergraph datasets.

- Motivated by recursively-structured datasets, we propose a novel extension of MPNN, termed MPNN-R (MPNN-Recursive) and show its effectiveness on real-world datasets.

## 2 Related Work

In this section, we discuss related work. In particular, we discuss relevant work on MPNNs, and their explorations on multi-relational graphs (including on the closely related multiplex networks, and heterogeneous graphs), and hypergraphs.

### 2.1 Message-Passing Neural Networks (MPNNs)

MPNNs were originally proposed as a framework for deep learning on graphs [25]. MPNN has inspired the current state-of-the-art techniques for graph representation learning (GRL). The reader is referred to comprehensive reviews [10, 7, 87] and extensive surveys [31, 90] on this topic of GRL. The message-passing operation in the MPNN framework can be viewed as recursive neighbourhood aggregation, where local neighbourhood messages are aggregated and passed on to neighbouring vertices [21]. Notable instances of the MPNN framework include popular graph neural networks such as *Graph Convolutional Networks* (GCNs) [43], *ChebNet* [17], *GraphSAGE* [30], *Graph Attention Networks* [68], *Neural Fingerprints* [18], *Gated Graph Sequence Neural Networks* (GGNN) [49], *Graph Isomorphism Networks* [76], etc. GNNs (and MPNNs) came into existence thanks to two seminal publications on convolutional [11] and recurrent [60] neural networks on graphs. The MPNN framework has been extended to multi-relational graphs in several ways which we discuss next.

### 2.2 MPNNs on Multi-Relational Graphs

The earliest attempts at extending GNNs to multi-relational graphs propose relation-specific parameters and include relational GGNN [29], and relational GCNs [61, 64]. Recent attempts include learning *relation-specific weights* [62], *neighbour and logic-based attention* [71], *triple-based attention* [56], *hierarchical attention* [91], and *relation and neighbour-based attention* [53]. The state-of-the art MPNN models include aggregating neighbour embeddings through *relation embedding composition* [67, 80, 88], *dynamically pruned MPNN on input-dependent subgraphs* [77], and *meta learning*

[3]. GCNs with direction-specific parameters have become quite popular approximations in natural language processing on syntactic multi-relational graphs [54, 6, 66].

**Multiplex Networks:** Another closely related line of research extends MPNN to *multiplex networks* or *multi-view* networks. Attempts along this line include effectively combining the Laplacians of the multiple views [41, 48], a *self-attention-based* approach [12], and an unsupervised embedding method [57] that extends Deep Graph Infomax [69].

**Heterogeneous Graphs:** A related line of research is MPNN extension to *heterogeneous graphs* in which vertices (and edges) are typed. Ideas successfully used along this line include *typed attention* [50], *neighbour attention* [84], *vertex-level and semantic-level attention* [73], *vertex, edge-type dependent parameterisation* [34] *metapath-based aggregation* [22], collective classification [95], and meta learning [35]. MPNN can also handle *edge features*[26, 15, 72].

### 2.3 MPNNs on Hypergraphs

Hypergraphs are challenging data structures as they encode relationships going beyond pairwise associations [93]. *Hypergraph Neural Networks* [20, 5] approximate the hypergraph by its clique expansion [1] and apply traditional graph-based deep approaches such as GCNs [14, 82, 36] on it. The clique expansion has been used subsequently in label propagation network [89], hyperedge prediction [8], and at least a couple of neural hypergraph construction methods [37, 38].

*Hypergraph Convolutional Network* [78] uses the mediator-based hypergraph Laplacian [13] which is one on of many non-linear Laplacians [32, 83, 47, 44, 46] and shows improvements on hypergraphs with noisy hyperedges. *Powerset Convolutional Networks* [75] uses tools from discrete signal processing to principally define convolution on set functions. *Hyper-SAGNN* [86] is a self-attention-based approach for hyperedge prediction [2]. HGCRNN [81] is proposed for temporal hypergraphs.

**Our contributions:** In all the publications that we have seen above, (hyper)edges *do not* participate as vertices in other (hyper)edges *recursively* and also *are not multi-relational ordered*. Our contributions are precisely to address these limitations of exisiting MPNN-based approaches.

## 3 Generalising MPNN to Multi-Relational Ordered hypergraphs

In this section, we will first see all the notations used. We will then briefly see MPNN before our proposed unified MPNN for multi-relational ordered hypergraphs.

### 3.1 Notations

We represent a multi-relational ordered hypergraph as a quadruple, $\mathcal{H} = (\mathcal{V}, \mathcal{E}, \mathcal{P}, \mathcal{R})$. Here, $\mathcal{V}$ is a set of $n$ vertices. $\mathcal{E} = \{e_1, \cdots, e_m\}$ is a *multiset of hyperedges* with each hyperedge $e \in \mathcal{E}$ satisfying $e \subseteq V$. The notation $\mathcal{P}$ is used to denote $\mathcal{P} := \{P_e : e \in \mathcal{E}\}$ where each $P_e$ is a positional mapping $P_e : e \to \{1, \cdots, p\}$, with $p$ an integer and $p \leq n$. We note that there can exist integers $i, j$ with $1 \leq i \leq m$, $1 \leq j \leq m$, and $i \neq j$ such that $e_i = e_j$ but $P_{e_i} \neq P_{e_j}$ (e.g. directed graph containing a directed edge in both forward and backward directions).

$\mathcal{R} : (\mathcal{E} \times \mathcal{P}) \to \{1, \cdots, r\}$ is a relation mapping that maps each (hyperedge, position) pair to one of $r$ pre-defined relations. We see that there can exist $(e_i, P_{e_i})$ and $(e_j, P_{e_j})$ such that $e_i = e_j$ but $\mathcal{R}(e_i, P_{e_i}) \neq \mathcal{R}(e_j, P_{e_j})$ (e.g. SonOf and FatherOf relationships). For a vertex $v \in \mathcal{V}$, let us define $I_v := \{e \in \mathcal{E} : v \in e\}$ i.e. $I_v$ is the set of hyperedges *incident* on $v$.

We now briefly describe MPNN [25] on an undirected graph $G = (V, E)$ with $N_v$ denoting the neighbourhood of vertex $v \in V$, and $e_{vw}$ denoting the edge features of an edge $\{v, w\} \in V$.

### 3.2 Message Passing Neural Network (MPNN)

The forward propagation of an MPNN has two phases viz., 1) a message passing phase (which runs for $T$ steps), and 2) a readout phase. The message passing phase is defined in terms of the message function $M_t$ and the vertex update function $U_t$ where $t$ is the time step with $t = 1, \cdots, T$. The message and the readout phases are respectively of the form

Table 1: Different existing instantiations of G-MPNN on different structures.

| Structure Type | Max. edge size, $\max_{e \in \mathcal{E}} |e|$ | # relations, $r$ |
|---|:---:|:---:|
| Graphs [25, 43, 30, 18, 49, 76] | 2 | 1 |
| Multi-relational graphs [29, 61, 64, 71, 56, 91, 53, 67, 77] | 2 | $\geq 1$ |
| Multiplex networks [41, 48, 12, 57] | 2 | $\geq 1$ |
| Heterogeneous graphs [50, 84, 73, 22, 34] | 2 | $\geq 1$ |
| Hypergraphs [20, 78, 37, 75, 79] | $\geq 2$ | 1 |

$$\textit{Message:} \quad m_v^{t+1} = \sum_{u \in N_v} M_t\left(h_v^t, h_u^t, e_{vw}\right), \quad h_v^{t+1} = U_t\left(h_v^t, m_v^{t+1}\right)$$
$$\textit{Readout:} \quad \hat{y} = R\left(\left\{h_v^T : v \in V\right\}\right)$$
(1)

where $M_t, U_t$, and $R$ are all differentiable functions. We will now see our generalised MPNN framework for multi-relational ordered hypergraphs.

### 3.3 G-MPNN (Generalised-MPNN)

Motivated by rich structures in real-world data (please see related work), we propose a generalised message function for a multi-relational ordered hypergraph $\mathcal{H} = (\mathcal{V}, \mathcal{E}, \mathcal{R}, \mathcal{P})$. The key modification is in the message function and is as follows.

$$m_v^{t+1} = g\left(\left\{M_t\left(h_v^t, \{(w, h_w^t)\}_{w \in e-v}, \ \mathcal{R}(e, P_e), P_e\right)\right\}_{e \in I_v}\right)$$
(2)

where $g$ is any parameterised differentiable function (e.g. element-wise mean, max, sum), $h_v^t$ is the hidden representation of a vertex $v \in V$ at time step $t$. The vertex update function, and the readout phase remain the same as in Equation 1.

Several notable models in the literature that can be described using the generalised MPNNs of Equation 2. Before describing the specific models, we will first see the following proposition that makes MPNN on a heterogeneous graph a special case of our generalised framework.

**Proposition 1.** *Let $G = (V, E, S)$ be a heterogeneous graph with $V$ as a set of vertices, $E$ as a set of directed edges, and a function $S : V \to \{1, \cdots, s\}$ that maps each $v \in V$ to a type $S_v$ to one of $s$ pre-defined types. Any heterogeneous graph $G = (V, E, S)$ is a special $\mathcal{H} = (\mathcal{V}, \mathcal{E}, \mathcal{R}, \mathcal{P})$ with*

- $\mathcal{V} = V$, and $\mathcal{E} = \{\{u, v\} : (u, v) \in E\}$
- $P_e(u) = 1$, and $P_e(v) = 2$ for each $(u, v) \in E$ (and $e \in \mathcal{E}$).
- $\mathcal{R}(e, P_e) = (s-1) * S_u + S_v$ for each $e \in \mathcal{E}$

*Proof.* The key step in the proof is item 3 above where we map the relation associated with each (edge, direction) pair in terms of the vertex types. We note that if $G$ has $s$ vertex types then $\mathcal{H}$ would have $s * (s-1)$ relations. Each directed edge $(u, v) \in E$ of $G$ is mapped to an edge $e = \{u, v\} \in \mathcal{E}$ whose relation is $\mathcal{R}_e = (s-1) * S_u + S_v$ with the position map $P_e(u) = 1$ and $P_e(v) = 2$. $\square$

The proposition can be trivially adapted to undirected edges, and trivially extended to heterogeneous graph with *both* node and edge types. Because of the proposition, we note that MPNNs on heterogeneous graphs are a special case of G-MPNN.

Several notable existing methods in the literature that can be described using our G-MPNN framework as we can see in Table 1. The proofs of how they can be described are in the appendix.

**Overview.** We now proceed to address limitations of existing MPNNs. In section 4, we introduce recursive hypergraphs, and then extend MPNN to MPNN-R (MPNN-Recursive) for recursive hypergraphs. Then, in the next subsection, motivated by the strong inductive capability of MPNN [30], we propose a novel instantiation of G-MPNN (Generalised-MPNN) framework, and explore the unexplored problem of inductive vertex embedding in multi-relational ordered hypergraphs. We leave exploration of G-MPNN-R as a promising future direction.

### 3.4 Inductive Vertex Embedding with G-MPNN

Though there exist several published works for learning on multi-relational ordered hypergraphs [74, 19, 28, 59, 51, 27], none of them is an MPNN-based approach and they all are restricted to the transductive setting (i.e. assume that all vertices are present during training). StarE [23] is an MPNN-based approach for hyper-relational graphs which are very different from multi-relational ordered hypergraphs. Motivated by the strong inductive capability of MPNN [30, 65] (and hence G-MPNN), we explore the unexplored problem of inductive learning (i.e. inductive vertex embedding) on multi-relational ordered hypergraphs.

We propose the following simple function for $M_t$:

$$M_t\left(h_v^t, \{(w, h_w^t)\}_{w \in e-v}, \ \mathcal{R}(e, P_e), P_e\right) = \mathbf{r_{e,P_e}^t} * \mathbf{p_{e,v}^t} * \mathbf{h_v^t} * \prod_{\mathbf{w \in e-v}} \left(\mathbf{p_{e,w}^t} * \mathbf{h_w^t}\right) \quad (3)$$

where we use bold letters for embeddings. We use linear layers to transform relation, and position embeddings between time steps i.e. $\mathbf{r_{e,P_e}^{t+1}} = W_r^t \, \mathbf{r_{e,P_e}^t}$ and $\mathbf{p_{e,v}^{t+1}} = W_p^t \, \mathbf{p_{e,v}^t}$, $\quad \forall e \in \mathcal{E}, v \in \mathcal{V}$. Our proposed $M_t$ can be seen as a generalisation of the following existing formulations $\forall e \in \mathcal{E}, v \in \mathcal{V}$:

- M-DistMult [19] when the position embeddings, $\mathbf{p_{e,v}^t} = \mathbf{1}$
- Bilinear Graph Convolution message function [94] when $\mathbf{r_{e,P_e}^t} = \mathbf{1}$, $\mathbf{p_{e,v}^t} = \mathbf{1}$, $|e| = 2$
- CompGCN-DistMult message function [67] when $\mathbf{p_{e,v}^t} = \mathbf{1}$, $|e| = 2$

For each vertex, our proposed method learns a function on the vertex embeddings of the vertex's neighbourhood, and any available vertex features. This precisely what enables the inductiveness of the method. It is worth noting that if the hypergraph $\mathcal{H}$ is attributed i.e. all the vertices, whether seen or unseen during training, are associated with input features (such as entity descriptions), then we use Equation 3 to compute the hidden representations of unseen vertices at test time. It is to be noted that the form of $M_t$ is not restricted to element-wise product of Equation 3 but other operators such as concatenation, convolution, etc.

## 4 MPNN-R: Message Passing for Recursive Hypergraphs

In this section, we will first see definitions with notations used. We will then see MPNN-R to handle recursive hypergraphs [55, 39].

**Definition 1** (Depth $k$ powerset). *For a set $S$, let us use $\mathcal{S}(S)$ to denote the powerset of $S$ i.e. $\mathcal{S}(S) := \{\dot{S} : \varnothing \subseteq \dot{S} \subseteq S\}$. Then, the the depth $k$ powerset of $S$ is*

$$2^{S,k} := \mathcal{S}\left(\bigcup_{i=0}^{k} S_i\right), \text{ where } S_0 = S, \text{ and } S_i = \mathcal{S}\left(\bigcup_{j=0}^{i-1} S_j\right), \text{ for } i \geq 1 \quad (4)$$

Note that $2^{S,0} = \mathcal{S}(S)$ i.e. $2^{S,0}$ is the powerset of $S$. Figure 1 shows a toy example of a depth 5 hyperedge with words as vertices. The whole hyperedge is of depth 5, pink hyperedge is of depth 4, blue of depth 3, green of depth 2, red of depth 1, and black of depth 0.

**Definition 2** ($k$-Recursive hypergraph). *A pair $H = (V, E)$, where $V$ is a set of $n$ vertices, and $E \subseteq \left(2^{V,k} - \varnothing\right)$ is a set of recursive hyperedges.*

Note that a hypergraph in the traditional sense is a 0-recursive hypergraph. In addition to the 5-recursive hyperedge of Figure 1, the following is a simpler example of a 1-recursive hypergraph.

**Example 1.** *(Academic network) Let $V$ be a set of documents, $E = E_0 \cup E_1$ be the set of recursive hyperedges. $E_0$ contains depth $0$ hyperedges of co-citation relationships (all documents cited by a document belong to a hyperedge). $E_1$ contains depth $1$ hyperedges of co-authorship relationships (all documents co-authored by an author belong to a hyperedge).*

An author (depth 1 hyperedge) can be a co-author of, say, 10 documents. Each of these 10 documents represents a depth 0 hyperedge connecting all its cited documents.

## 4.1 Laplacian-based MPNN

One of the most popular (if not the most popular), widely used examples of MPNN on graphs is the Laplacian-based *convolutional neural network on graphs* [43, 17]. From a signal processing perspective, 2D convolutions are linear, shift-invariant (i.e. equivariant) functions on image grid data. By analogy, *graph convolution* on vertex-indexed signals, $x : \mathcal{V} \to \mathbb{R}; v \to x_v$, can be interpreted as a linear and equivariant operator with respect to (powers of) Laplacian shifts i.e. $L^i x$ where $L$ is the graph Laplacian [63, 11]. Graph convolutional network (GCN) [43] uses a linear function of $L$ (the graph Laplacian) for the (layer-wise) convolution operation .

## 4.2 Laplacian of Recursive Hypergraph

Inspired by GCN, we propose a Laplacian-based MPNN-R for recursive hypergraphs. To be able to define such an MPNN, we need a notion of Laplacian for recursive hypergraphs. Our idea crucially relies on the relationship between a Laplacian matrix and incidence matrix.

**Definition 3** (Incidence structure [4]). *A set of objects (e.g. vertices) together with certain incidence relations between these objects (e.g. hyperedges) is an incidence structure.*

An incidence matrix is a rectangular matrix that shows the relationship between two classes of objects (e.g. vertices and hyperedges) in an incidence structure. It is typically represented by a rectangular $|V| \times |E|$ matrix where $V$ and $E$ are two sets. Now, we state a well-known relationship between the incidence and the Laplacian matrices

**Theorem 1.** *Let $H = (V, E)$ be an incidence structure and $\mathcal{I}$ be an (arbitrarily chosen) incidence matrix and Laplacian $L$. Then $\mathcal{I}\mathcal{I}^T = L$*

The proof is seen in standard literature (e.g. a book [24]). The key idea of our proposed MPNN-R is to appropriately choose an incidence matrix $\mathcal{I}$ for the input recursive hypergraph, and then use the resulting Laplacian in a Laplacian-based MPNN.

**Incidence Matrix of Recursive Hypergraph.** Recall that in a recursive hypergraph $H = (V, E)$, hyperedges can act as vertices in other hyperedges. Hence, we define the new vertex set $U = V \cup E$ with the same hyperedge set $E$. The incidence matrix $\mathcal{I}$ is hence a $|U| \times |E|$ matrix where each entry $\mathcal{I}_{ue}$ indicates the "strength" of the membership of $u \in U$ in the hyepredge $e \in E$. Note that for two different hyperedges $e_1 \in E$ and $e_2 \in E$, the strengths might be different i.e. $\mathcal{I}_{ue_1} \neq \mathcal{I}_{ue_2}$. This can be seen as a generalisation of hyperedge-dependent vertex weights [16, 45] to recursive hypergraphs. Hyperedge-dependent vertex weights are known to utilise higher-order relationships in hypergraphs.

## 4.3 Laplacian-based MPNN-R for Recursive Hypergraph

From Theorem 1, we can now compute the Laplacian matrix of the recursive hypergraph as $L = \mathcal{I}\mathcal{I}^T$ (where $\mathcal{I}$ is obtained as above). Our MPNN-R then takes the following form for $v \in U$:

$$m_v^{t+1} = g\left(\left\{ M_t\left(h_v^t, h_u^t, e_{vw}\right)\right\}_{u \in N_v}\right), \quad h_v^{t+1} = U_t\left(h_v^t, m_v^{t+1}\right)$$

$$M_t\left(h_v^t, h_u^t, e_{uv}\right) = L_{vu}h_u^t, \quad U_t\left(h_v^t, m_v^{t+1}\right) = \sigma\left(\left(W^t\right)^T m_v^{t+1}\right) \tag{5}$$

We define the "neighbourhood" of $v$ as those vertices whose entries in the Laplacian matrix corresponding to row of vertex $v$ are non-zero. It is easy to see that Laplacian-based MPNNs for graphs such as GCN [43] and hypergraphs such as HGNN [20] are examples of our MPNN-R (for special incidence matrices) because they essentially work on 0-recursive (hyper)graphs.

Table 2: Results of SSL experiments. We report mean test error $\pm$ standard deviation (lower is better) over 100 train-test splits. Please refer to section 5.1 for details.

| Method | Cora | DBLP | ACM | arXiv |
|---|---|---|---|---|
| MLP | $42.14 \pm 1.8$ | $37.72 \pm 1.9$ | $34.54 \pm 1.5$ | $39.76 \pm 2.3$ |
| HGNN | $32.41 \pm 1.8$ | $24.98 \pm 2.0$ | $27.56 \pm 1.5$ | $31.32 \pm 1.7$ |
| HyperGCN | $32.37 \pm 1.7$ | $24.76 \pm 2.2$ | $27.12 \pm 1.3$ | $31.25 \pm 1.8$ |
| HetGNN | $27.45 \pm 1.3$ | $22.15 \pm 2.0$ | $23.43 \pm 1.9$ | $25.55 \pm 2.0$ |
| HAN | $27.24 \pm 1.9$ | $22.18 \pm 1.4$ | $23.21 \pm 2.1$ | $25.02 \pm 2.2$ |
| MAGNN | $26.78 \pm 1.5$ | $\mathbf{21.68 \pm 1.8}$ | $22.29 \pm 1.9$ | $24.23 \pm 1.8$ |
| **MPNN-R (ours)** | $\mathbf{25.34 \pm 1.5}$ | $\mathbf{21.45 \pm 1.7}$ | $\mathbf{20.32 \pm 2.1}$ | $\mathbf{22.34 \pm 1.7}$ |

# 5 Evaluation

We evaluate the performance of the proposed methods in a number of experiments. Following standard practices of prior works, we focus on the following two most popular tasks.

## 5.1 MPNN-R: Semi-Supervised Vertex Classification

We evaluate MPNN-R on the task of semi-supervised classification of documents in academic network datasets. The input is a 1-recursive hypergraph with documents as vertices, words as features (bag-of-words), authors as depth 1-hyperedges, and references in documents as depth 0 hyperedges. The task is multi-class classification of documents given the input recursive hypergraph, and a small fraction of labelled documents in the dataset (we call the fraction label rate, please see label rate and other details in dataset statistics table in the appendix.

**Experimental Set-up.**  To contextualise the experimental results on the datasets, we compare against the following four baselines:

- **Multi-layer Perceptron (MLP):** We use the initial features into a simple 2-layer feed-forward neural network to predict the class labels of instances. This baseline ignores the recursive hypergraph structure.

- **HyperGraph Neural Network (HGNN [20]):** HGNN approximates the input hypergraph by introducing pairwise connections among *all vertex pairs* in each hyperedge [92]. This is exactly the same as a straightforward extension of GCN [43] to hypergraphs. We use all the depth 0-hyperedges to form the input hypergraph for HGNN.

- **HyperGraph Convolutional Network (HyperGCN [78]):** HyperGCN uses the mediator expansion [13] to approximate the input hypergraph into a graph. As with HGNN, we use all the depth 0-hyperedges to form the input hypergraph for HyperGCN.

- **Heterogeneous Graph Neural Network (HetGNN):** We compare against a GNN-based method on heterogeneous graphs [84]. We treat instances, depth 0-hyperedges, and depth 1-hyperedges as three different types of vertices in a heterogeneous graph. We connect an instance, and a depth 0-hyperedge if the instance belongs to the hyperedge. We connect a depth 0, and a depth 1-hyperedge if the depth 0-hyperedge belongs to the depth 1-hyperedge.

- **Heterogeneous Attention Network (HAN) [73]**: This is another GNN-based method on heterogeneous graphs that uses an attention-based formulation.

- **Metapath-Aggregated Graph Neural Network (MAGNN) [22]**: This is a recent GNN-based method on heterogeneous graphs that exploits metapaths (schemas) for neighbourhood aggregation.

**Model Details:**  We use a 2-layer MPNN-R of Equation 5 with ReLU as the non-linear activation function. We use 1024-dimensional hidden embeddings with $c$-dimensional output embeddings where $c$ is the number of classes as shown in dataset statistics table in the appendix. We set the hyperedge-dependent vertex weights to one for all vertices i.e. $\mathcal{I}_{ue} = 1$ if $u \in e$. Please see the appendix for comparison with different hyperedge-dependent vertex weights. We use the popularly-used (symmetrically-normalised) mean aggregator to aggregate messages from the neighbourhood i.e.

Table 3: Results of link prediction experiments. We report MRR, Hits@1, and Hits@3 (higher is better) on held-out test sets. Please refer to section 5.2 for details.

| Method | WP-IND | | | JF-IND | | | MFB-IND | | |
|---|---|---|---|---|---|---|---|---|---|
| | MRR | Hits@1 | Hits@3 | MRR | Hits@1 | Hits@3 | MRR | Hits@1 | Hits@3 |
| HGNN | 0.072 | 0.045 | 0.112 | 0.102 | 0.086 | 0.128 | 0.121 | 0.076 | 0.114 |
| HyperGCN | 0.075 | 0.049 | 0.111 | 0.099 | 0.088 | 0.133 | 0.118 | 0.074 | 0.117 |
| **G-MPNN-sum (ours)** | 0.177 | 0.108 | 0.191 | **0.219** | **0.155** | 0.236 | 0.124 | 0.071 | 0.123 |
| **G-MPNN-mean (ours)** | 0.153 | 0.096 | 0.145 | 0.112 | 0.039 | 0.116 | 0.241 | 0.162 | 0.257 |
| **G-MPNN-max (ours)** | **0.200** | **0.125** | **0.214** | 0.216 | 0.147 | **0.240** | **0.268** | **0.191** | **0.283** |

given a set of hidden embeddings denoted by $h$ we use the following function in Equation 5

$$g\Big(\{h_u\}_{u \in N_v}\Big) = \frac{1}{\sqrt{|N_v| \cdot |N_u|}} \sum_{u \in N_v} h_u$$

We found that sum and max aggregators perform comparably to the mean aggregator. Please see the appendix for detailed ablation studies. We train our MPNN-R with cross entropy loss function on the labelled vertices following standard practice of prior works [43, 20, 78]. All models are implemented in PyTorch [58] using the Adam optimiser [42].

**Experimental Results:** We take extensive steps to avoid any kind of bias in our results. We optimise all hyperparameters of all baselines and our method using grid search. Please see the appendix for more details on the hyperparameters used. Table 2 shows the results of semi-supervised vertex classification on real-world datasets. We report mean errors and their standard deviations (lower is better) of the test splits over 100 different train-test splits.

**Discussion:** As we can see from Table 2, MLP is least effective on all datasets. This shows that the input recursive hypergraph is informative in classifying vertices. We can also observe that HetGNN that uses both depth 0 and depth 1-hyperedges is consistently superior to HGNN and HyperGCN both of which use only the depth 0-hyperedges. Finally, our proposed method MPNN-R is able to consistently outperform all the baselines including HetGNN. We believe this is because of the principled integration of depth 0, depth 1-hyperedges, and also relationships among them. Finally we performed a statistical test to validate the significance of the results. Based on a Welch t-test, the p-value on DBLP is 0.35, and a small p-value on the other three (less than 0.0005).

## 5.2 G-MPNN: Link Prediction

We evaluate G-MPNN on the task of link prediction on $N$-ary relational facts. Since the inductive version (embedding unseen entities at test time) is an unexplored task, we create datasets from existing (transductive) datasets. Details of the dataset construction and statistics are in the appendix.

**Inductive Vertex Embedding:** We concatenate the neighbour embedding inspired with Equation 3 and the self features of the vertices. In other words, if $\mathbf{seen(e)}$ is the set of vertices in hyperedge $e$ seen during training then for an unseen vertex $v \in e$, we modify Equation 3 to

$$M_t\Big(h_v^t, \big\{(w, h_w^t)\big\}_{w \in e-v}, \ \mathcal{R}(e, P_e), P_e\Big) = \mathbf{r_{e,P_e}^t} * \mathbf{\Psi_{w \in seen(e)}}\Big(\mathbf{p_{e,w}^t} * \mathbf{h_w^t}\Big) \qquad (6)$$

where $\Psi_{w \in seen(e)}$ is a randomly chosen vertex $w$ from $seen(e)$

**Objective Function:** As the input hypergraph has only positive hyperedges, we produce a set of $\eta$ negative hyperedges through a standard contrastive generation procedure [9]. The score, $\phi(e)$, of a hyperedge, $e$, after obtaining G-MPNN vertex hidden representations, is computed using generalised bilinear scoring for $N$-ary facts [19] i.e. $\phi(e) = \mathbf{r_{e,P_e}^T} * \prod_{\mathbf{w \in e}} \big(\mathbf{p_{e,w}^T} * \mathbf{h_w^T}\big)$ where $T$ is the total number of G-MPNN time steps. We use a combination of softmax and negative log likelihood loss which is shown to be effective for link prediction [40].

$$\mathcal{L} = \sum_{e \in \mathcal{E}} -\log\Bigg(\frac{exp\big(\phi(e)\big)}{exp\big(\phi(e)\big) + \sum_{e' \in neg(e)} exp\big(\phi(e)\big)}\Bigg)$$

**Experimental Set-up:** To contextualise the experimental results on the datasets, we compare against two baselines - HGNN [20] and HyperGCN [78] adapted to the problem of link prediction. Both HGNN and HyperGCN were originally proposed for vertex-level tasks on hypergraphs but we adapt them to link prediction task by removing the relational and ordering information from the multi-relational ordered hypergraph. We use the hidden representations obtained from these models as embeddings for the scoring function $\phi(e)$.

**Model Details:** We use one layer of generalised message passing We use 150-dimensional hidden units. We use three popular forms for the function $g$ viz., sum, mean, and max [70] which we call G-MPNN-sum, G-MPNN-mean, and G-MPNN-max respectively. All models are implemented in PyTorch [58] using Adam [42].

**Metrics for Evaluation:** We rely on two popular ranking metrics viz. Hits@$k$, and Mean Reciprocal Rank (MRR). Both these metrics rank a test hyperedge $e$ within a set of corrupted (negative) hyperedges. Since the number of positions is small, for each $(e, P_e)$ pair and for each position, we generate $|\mathcal{V}| - 1$ corrupted hyperedges by replacing the vertex $v \in e$ at position $i$ with each of the entities in $\mathcal{V} - v$. We rank the positive $e$ based on score $\phi(e)$. We compute MRR as mean of the sum (over all positions) of the reciprocals of the rank of $e$. Hits@$k$ is the proportion of hyperedges that rank among top $k$ in corrputed sets.

**Experimental Results and Discussion:** The results of link prediction are shown in Table 3. Firstly, our proposed methods are more effective than HGNN and HyperGCN. We believe this is because the two baselines do not exploit the positional and relational information in the hypergraph. We have conducted ablation studies by removing positional and relational information and the results are in the appendix. We have also conducted experiments on transductive datasets (results in the appendix).

## Conclusion

In this work, we have unified existing MPNN approaches proposed on a wide range of networks such as multi-relational graphs, hypergraphs, heterogeneous graphs, etc. Our unified framework, G-MPNN, has attractive properties including strong inductive capability on multi-relational ordered hypergraphs. We have also proposed a novel framework, MPNN-R, on recursive hypergraphs. Future possibilities include exploiting relational and positional information in recursive hypergraphs (see Figure 1) for natural language processing tasks. Another interesting direction is to extend recent subgraph reasoning methods [85, 65, 52] for inductive vertex embedding on multi-relational ordered hypergraphs. Scaling our methods for large datasets [33] is also an interesting direction.

## Broader Impact

Message Passing Neural Networks (MPNNs) are a framework for deep learning on graph structured data. Graph structures are universal and very generic structures commonly seen in various forms in computer vision, natural language processing, recommender systems, traffic prediction, generative models, and many more. Graphs can have many variations such as multi-relational, heterogeneous, hypergraphs, etc.

Our research in this paper unifies several existing MPNN methods on these variations. While we show how our research could be used for academic networks, and factual knowledge, it opens up many more possibilities in natural language processing (NLP). We see opportunities for research applying our work for beneficial puroposes, such as investigating whether we could improve performance of NLP tasks such as reading comprehension, relation extraction, machine translation, and many more.

Potentially hazardous applications include trying to predict criminality or credit from social networks. Such applications may reproduce and exacerbate bias and readers of the paper should be aware that the presented model should not applied naively to such tasks.

## Funding Disclosure and Acknowledgements

This work is supported by the Ministry of Human Resource Development (Government of India). We give credits to Google Images and Wikipedia for the pictures used in the paper.

## Footnotes

[1] https://github.com/naganandy/G-MPNN-R

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
