[Supplementary Material]

# Neural Message Passing for Multi-Relational Ordered and Recursive Hypergraphs (Appendix)

**Contents.** The sections are organised as follows.

## 1 Existing MPNNs as Special Cases of G-MPNN

We rewrite the formulation of G-MPNN for improved readability.

$$m_v^{t+1} = g\left( \left\{ M_t\left( h_v^t, \left\{ (w, h_w^t) \right\}_{w \in e-v}, \ \mathcal{R}(e, P_e), P_e \right) \right\}_{e \in I_v} \right) \tag{1}$$

### 1.1 MPNNs on Multi-relational graphs

To the best of our knowledge, there are no published works on MPNNs for knowledge hypergraphs. Hence, MPNN on multi-relational graphs takes the following form:

$$m_v^{t+1} = g\left( \left\{ M_t\left( h_v^t, h_w^t, \ \mathcal{R}(e, P_e), P_e \right) \right\}_{e \in N_v} \right) \tag{2}$$

where $N_v$ is the multi-relational neighbourhood of $v$. A standard practice in embedding multi-relational graphs has been to introduce an inverse relation for each existing relation. In other words, for each $(s, r, o)$ triple in the multi-relational graph, we add $(o, r^{-1}, s)$ to the set of existing triples. Under this setting, it is redundant and not necessary to assume positional information given be $P_e$. Hence, MPNN on multi-relational graphs takes the following simpler form:

$$m_v^{t+1} = g\left( \left\{ M_t\left( h_v^t, h_w^t, \ R_e \right) \right\}_{e \in N_v} \right) \tag{3}$$

where $R_e$ is the relation of the edge (triple) connecting $v$ and $w$. We assume, without loss of generality, that $v$ is the object of the triple $(w, R_e, v)$.

**Relational GCN [17, 12]**   R-GCN uses relation-specific filters/weight matrices for aggregation i.e. $M_t\left( h_v^t, h_w^t, R_e \right) = W_{R_e} h_w^t$.

**Structure-Aware Convolutional Network [18]**   SACN uses relation-specific scalar-valued i.e. $M_t\left( h_v^t, h_w^t, R_e \right) = \alpha_{R_e} h_w^t$

**Composition-based Multi-Relational Graph Convolutional Networks [23]**   CompGCN uses relation-specific embeddings in a composition operator $\Phi$ i.e. $M_t\left( h_v^t, h_w^t, R_e \right) = \Phi(R_e, h_w^t)$

## 1.2   MPNNs on Hypergraphs

Existing hypergraph MPNNs are mono relational i.e. fall under the following formulation:

$$m_v^{t+1} = g\left( \left\{ M_t\left( h_v^t, \{ (w, h_w^t) \}_{w \in e - v} \right) \right\}_{e \in I_v} \right) \tag{4}$$

**Hypergraph Neural Networks [10, 14]**   use the clique reduction of the hypergraph [29] to graph. Hence $M_t\left( h_v^t, \{ (w, h_w^t) \}_{w \in e - v} \right) = \sum_{w \in e - v} h_w^t$.

**Hypergraph Convolutional Network [26]**   uses the mediator expansion [5] to approximate the hypergraph to graph. Each hyperedge is approximated by a tripartite subgraph as follows. Consider the maximally disparate vertices of the hyperedge $e$ i.e. the supremum $s_e$, and the infimum $i_e$ given by $s_e, i_e = \arg\max_{j,k \in e} |h_j - h_k|^2$. Let the vertices $M_e = \{ m \in e : \ m \neq s_e, m \neq i_e \}$ represent the set of mediators. Then the tripartite graph is the graph with $\{s_e\}$, $\{i_e\}$, and $M_e$ as the three partitions. The message function is thus

$$M_t\left( h_v^t, \{ (w, h_w^t) \}_{w \in e - v} \right) = \begin{cases} \sum_{w \in e - v} h_w^t & \text{if } w \in \{s_e, i_e\} \\ h_{s_e}^t + h_{i_e}^t & \text{if } w \in M_e \end{cases}$$

**PowerSet Convolutional Network [25]**   Powerset convolution is defined on hyperedges. However, we can go to the dual hypergraph (where vertices become hyperedges and hyperedges become vertices) and pose PCN as a special instance of our framework. In particular, the first-order PCN can be seen as $M_t\left( h_v^t, \{ (w, h_w^t) \}_{w \in e - v} \right) = \sum_{w \in \mathcal{N}(v)} h_w^t$ where $\mathcal{N}(v)$ is defined as the set of those vertices such that $|I(v) \setminus I(w)| = 1$

## 1.3   MPNNs on Heterogeneous networks

The proofs are trivial and follow in a straightforward way from the following proposition (restated for completeness):

**Proposition 1.** *Let $G = (V, E, S)$ be a heterogeneous graph with $V$ as a set of vertices, $E$ as a set of directed edges, and a function $S : V \to \{1, \cdots, s\}$ that maps each $v \in V$ to a type $S_v$ to one of $s$ pre-defined types. Any heterogeneous graph $G = (V, E, S)$ is a special $\mathcal{H} = (\mathcal{V}, \mathcal{E}, \mathcal{R}, \mathcal{P})$ with*

- $\mathcal{V} = V$, and $\mathcal{E} = \{\{u, v\} : (u, v) \in E\}$

- $P_e(u) = 1$, and $P_e(v) = 2$ for each $(u, v) \in E$ (and $e \in \mathcal{E}$).

- $\mathcal{R}(e, P_e) = (s - 1) * S_u + S_v$ for each $e \in \mathcal{E}$

Similarly, it is also trivial to instantiate MPNNs on multiplex networks in our framework.

## 2 Appendix for MPNN-R

### 2.1 Algorithm

In this section, we describe the generation of vertex embeddings through with the help of an algorithm MPNN-R. Let $H = (V, E)$ be a recursive hypergraph, where $V$ is a set of $n$ vertices, and $E \subseteq \left(2^{V,k} - \varnothing\right)$ is a set of recursive hyperedges.

**Incidence Matrix of Recursive Hypergraph.** Recall that in a recursive hypergraph $H = (V, E)$, hyperedges can act as vertices in other hyperedges. Hence, we define the new vertex set $U = V \cup E$ with the same hyperedge set $E$. The incidence matrix $\mathcal{I}$ is hence a $|U| \times |E|$ matrix where each entry $\mathcal{I}_{ue}$ indicates the "strength" of the membership of $u \in U$ in the hyepredge $e \in E$. Note that for two different hyperedges $e_1 \in E$ and $e_2 \in E$, the strengths might be different i.e. $\mathcal{I}_{ue_1} \neq \mathcal{I}_{ue_2}$. This can be seen as a generalisation of hyperedge-dependent vertex weights [6, 16] to recursive hypergraphs. Hyperedge-dependent vertex weights are known to utilise higher-order relationships in hypergraphs.

---

**Algorithm 1:** Algorithm for MPNN-R vertex embedding generation

---

**Input** : Recursive Hypergraph $H = (V, E)$; input features $\{\mathbf{x}_v, \forall v \in V\}$; depth $L$; weight matrices $\mathbf{W}^l, \forall l \in \{1, ..., L\}$; non-linearity $\eta$; differentiable aggregator functions $\text{AGG}_l, \forall l \in \{1, ..., L\}$;

**Output :** Vector representations $\mathbf{z}_v$ for all $v \in \mathcal{V}$

1 $\mathbf{h}_v^0 \leftarrow \mathbf{x}_v, \forall v \in \mathcal{V}$ ;
2 Include hyperedges in the set of vertices $\mathcal{V} = V \cup E$ ;
3 Obtain incidence matrix $\mathcal{I}$ from $H$ and $\mathcal{V}$ as described ;
4 Compute the neighbourhood function $\mathcal{N} : v \to 2^{\mathcal{V}}$ from the Laplacian matrix ;
5 **for** $k = 1...L$ **do**
6     **for** $v \in \mathcal{V}$ **do**
7         $\mathbf{h}_v^l \leftarrow \eta \left( \mathbf{W}^l \cdot \text{AGG}_l(\{\mathbf{h}_u^{l-1}, \forall u \in \mathcal{N}(v)\}) \right)$
8     **end**
9     $\mathbf{h}_v^l \leftarrow \mathbf{h}_v^l / \|\mathbf{h}_v^l\|_2, \forall v \in \mathcal{V}$
10 **end**
11 $\mathbf{z}_v \leftarrow \mathbf{h}_v^L, \forall v \in \mathcal{V}$

---

### 2.2 Comparison with Sum and Max Aggregators

We compare other aggregators with the mean aggregator used for MPNN-R in the main contents of oure work. The other aggregators perform very similar to the mean as shown in Table 1.

Table 1: Comparison with Sum and Max Aggregators.

| Method | Cora | DBLP | ACM | arXiv |
|---|---|---|---|---|
| MPNN-R-sum | $25.30 \pm 1.6$ | $21.54 \pm 1.5$ | $20.25 \pm 1.7$ | $22.40 \pm 1.4$ |
| MPNN-R-max | $25.41 \pm 1.6$ | $21.44 \pm 1.3$ | $20.35 \pm 2.0$ | $22.36 \pm 1.6$ |
| MPNN-R-mean | $25.34 \pm 1.5$ | $21.45 \pm 1.7$ | $20.32 \pm 2.1$ | $22.34 \pm 1.7$ |

## 2.3 Hyperedge-dependent Vertex Weights

Recall that the incidence matrix of the input recursive hypergraph is $\mathcal{I}$, a $|U| \times |E|$ matrix where each entry $\mathcal{I}_{ue}$ indicates the "strength" of the membership of $u \in U$ in the hyepredge $e \in E$. We use prior heuristic knowledge in academic networks e.g. first authors are likely to be very focused, etc. We set the strength of author-dependent vertex weight to a hyperparameter $\eta$ if the author is the first author (else one). Table 2 shows improvements for the mean aggregator on all datasets.

Table 2: $\eta$ is the author-dependent vertex weight if the author is the first author. .

| Method | Cora | DBLP | ACM | arXiv |
|---|---|---|---|---|
| MPNN-R | $25.34 \pm 1.5$ | $21.45 \pm 1.7$ | $20.32 \pm 2.1$ | $22.34 \pm 1.7$ |
| MPNN-R ($\eta = 2$) | $25.30 \pm 1.4$ | $\mathbf{21.33 \pm 1.5}$ | $20.28 \pm 1.8$ | $22.35 \pm 1.5$ |
| MPNN-R ($\eta = 4$) | $\mathbf{25.26 \pm 1.5}$ | $21.38 \pm 1.6$ | $\mathbf{20.19 \pm 1.8}$ | $\mathbf{22.27 \pm 1.6}$ |
| MPNN-R ($\eta = 8$) | $25.37 \pm 1.8$ | $21.47 \pm 1.3$ | $20.33 \pm 2.5$ | $22.29 \pm 1.8$ |

## 2.4 Dataset Construction

We briefly provide details of how we construct recursive hypergraph datasets.

**Cora:** We used the author data[1] to get the co-authorship relationships for cora. We use cocitation relationships from [2].

**DBLP, ACM, arXiv:** We obtained the full dblp [3] and ACM [4] datasets from a published work [20]. We obtained arXiv [5] from another work [7]. We used conference categories from Wikipedia [6] as a guide to curate our data. Speifically, we defined a set of conference categories (classes) as "algorithms", "database", "datamining", "intelligence", "vision", etc.. We extracted authors and publications from these conferences to get the recursive hypergraph.

## 2.5 Dataset Statistics

Table 3: Dataset statistics in the experiments for MPNN-R.

| Dataset | # vertices | # features | # depth 0-hyperedges | # depth 1-hyperedges | # classes | Label rate |
|---|---|---|---|---|---|---|
| Cora | $2,708$ | $1,433$ | $1,579$ | $1,072$ | $7$ | $0.052$ |
| DBLP | $52,040$ | $869$ | $20,988$ | $21,777$ | $5$ | $0.050$ |
| ACM | $100,376$ | $4,684$ | $39,266$ | $42,656$ | $3$ | $0.100$ |
| ArXiv | $63,660$ | $410$ | $32,856$ | $46,618$ | $3$ | $0.001$ |

## 2.6 Varying Labelled Data

We conduct experiments by varying labelled data on the arXiv dataset. We use the mean aggregator. Table 4 shows superior performance on $1\%, 3\%, 5\%, 10\%, 20\%$ labelled datasets.

## 2.7 Computational Complexity and Hyperparameters

Let $n_0$ be the number of depth 0 hyperedges, $n_1$ be the number of depth 1 hyperedges, and $n$ be the number of vertices. Let $N = n_0 + n_1 + n$ and $d$ be the number of hidden units in the hidden layer. Assuming that real-world hypergraphs are sparse, the complexity of MPNN-R (for depth 1-recursive hypergraphs) is $O(Nd)$.

Table 4: Results on *arXiv* dataset. $100*$ Mean squared error $\pm$ standard deviation (lower is better) over 10 different train-test splits.

| Model | 1% | 3% | 5% | 10% | 20% |
|---|---|---|---|---|---|
| HGNN | $34.78 \pm 1.6$ | $32.12 \pm 1.8$ | $31.32 \pm 1.7$ | $31.75 \pm 1.6$ | $30.65 \pm 1.7$ |
| HyperGCN | $34.80 \pm 1.5$ | $32.15 \pm 1.6$ | $31.25 \pm 1.8$ | $31.76 \pm 1.5$ | $30.60 \pm 1.6$ |
| HetGNN | $28.89 \pm 1.9$ | $25.02 \pm 1.8$ | $25.55 \pm 2.0$ | $26.06 \pm 1.8$ | $25.23 \pm 1.9$ |
| **MPNN-R (Ours)** | $\mathbf{25.06 \pm 1.8}$ | $\mathbf{22.01 \pm 1.4}$ | $\mathbf{22.34 \pm 1.7}$ | $\mathbf{22.87 \pm 1.6}$ | $\mathbf{21.96 \pm 2.2}$ |

Experiments were run for 200 epochs on a GTX 1080 Ti with 12 GB RAM. The Adam optimiser was used with a learning rate of $0.01$, L2 penalty of $5e^{-4}$. Following standard practice, the size of $d$ was set to 32. The model was evaluated on the validation set and saved three epochs with the best performing checkpoint used for testing.

## 3 Appendix for G-MPNN

### 3.1 Algorithm

$$M_t\left(h_v^l, \left\{(w, h_w^l)\right\}_{w \in e-v}, \ \mathcal{R}(e, P_e), P_e\right) = \mathbf{r_{e,P_e}^l} * \mathbf{p_{e,v}^l} * \mathbf{h_v^l} * \prod_{\mathbf{w} \in \mathbf{e-v}} \left(\mathbf{p_{e,w}^l} * \mathbf{h_w^l}\right) \quad (5)$$

---
**Algorithm 2:** Algorithm for G-MPNN vertex embedding generation

**Input** : Multi-Relational Ordered Hypergraph $\mathcal{H} = (\mathcal{V}, \mathcal{E}, \mathcal{P}, \mathcal{R})$; input features $\{\mathbf{x}_v, \forall v \in V\}$; depth $L$; weight matrices $\mathbf{W}^l, \forall l \in \{1, ..., L\}$; non-linearity $\eta$; differentiable aggregator functions $\text{AGG}_l, \forall l \in \{1, ..., L\}$;

**Output :** Vector representations $\mathbf{z}_v$ for all $v \in \mathcal{V}$

1 $\mathbf{h}_v^0 \leftarrow \mathbf{x}_v, \forall v \in \mathcal{V}$ ;
2 Include hyperedges in the set of vertices $\mathcal{V} = V \cup E$ ;
3 Obtain incidence matrix $\mathcal{I}$ from $H$ and $\mathcal{V}$ as described ;
4 Compute the neighbourhood function $\mathcal{N} : v \to 2^{\mathcal{V}}$ from the Laplacian matrix ;
5 **for** $k = 1...L$ **do**
6     **for** $v \in \mathcal{V}$ **do**
7        $\mathbf{h}_v^l \leftarrow \eta\left(\mathbf{W}^l \cdot \text{AGG}_l(\{M_t\left(h_v^l, \left\{(w, h_w^l)\right\}_{w \in e-v}, \ \mathcal{R}(e, P_e), P_e\right)\})\right)$
8     **end**
9     $\mathbf{h}_v^l \leftarrow \mathbf{h}_v^l / \|\mathbf{h}_v^l\|_2, \forall v \in \mathcal{V}$
10 **end**
11 $\mathbf{z}_v \leftarrow \mathbf{h}_v^L, \forall v \in \mathcal{V}$

---

### 3.2 Ablation Study

We perform ablation studies on WikiPeople dataset to verify that all the information used by our method is necessary to achieve the best performance. One set of ablated baselines neither uses the relational information nor the positional information (poisition/relation) embeddings are set to vectors of all ones). Another set of ablated baselines uses only the relational information while a third set uses only the positional information. Table 5 shows the results.

Table 5: Ablation Study on the MFB-IND dataset.

| Method | Relation | Position | MFB-IND | | |
|---|---|---|---|---|---|
| | | | MRR | Hits@1 | Hits@3 |
| **G-MPNN-mean** | $\times$ | $\times$ | 0.158 | 0.126 | 0.191 |
| **G-MPNN-max** | $\times$ | $\times$ | 0.163 | 0.125 | 0.185 |
| **G-MPNN-mean** | $\times$ | $\checkmark$ | 0.196 | 0.142 | 0.211 |
| **G-MPNN-max** | $\times$ | $\checkmark$ | 0.199 | 0.143 | 0.204 |
| **G-MPNN-mean** | $\checkmark$ | $\times$ | 0.209 | 0.152 | 0.220 |
| **G-MPNN-max** | $\checkmark$ | $\times$ | 0.203 | 0.162 | 0.213 |
| **G-MPNN-mean** | $\checkmark$ | $\checkmark$ | 0.241 | 0.162 | 0.257 |
| **G-MPNN-max** | $\checkmark$ | $\checkmark$ | **0.268** | **0.191** | **0.283** |

## 3.3 Dataset Construction

We constructed inductive datasets from existing transductive datasets (Wikipeople [11], JF17K, and M-FB15K [9]) We need test sets containing unseen entities (i.e. not seen during training). The steps taken are similar to the binary case [12, 24] and are as follows:

- Sample a fraction of the original test hyperedges to form a new test set $T$.

- Add all entities in $T$ to an auxiliary unseen set $U'$

- Remove entities in $U'$ which do not appear in any fact hyperedges in the training set to yield the final unseen entity set $U$

- Remove a fact hyperedge in $T$ if all entities in the hyperedge are seen in training

- Split the original training set into new training set and auxiliary set

- Add a fact hyperedge to the new training set if all entities in the hyperedge are seen in training

- Add the remaining facts in the original train set (i.e. hyperedges in which at least one entity is unseen) to the auxiliary set

- Filter out a fact hyperedge from validation if it contains at least one unseen entity We train all our methods and baselines on the new training set, optimise hyperparameters using the filtered validation set, and test on the methods on the new test set with the auxiliary set used as the signals for G-MPNN.

## 3.4 Dataset Statistics

Table 6: Dataset statistics in the experiments for G-MPNN.

| Dataset | # seen vertices | # train hyperedges | # unseen vertices | # relations | # features |
|---|---|---|---|---|---|
| WP-IND | $4,363$ | $4,139$ | 100 | 32 | 37 |
| JF-IND | $4,685$ | $6,167$ | 100 | 31 | 46 |
| MFB-IND | $3,283$ | $336,733$ | 500 | 12 | 25 |

## 3.5 Binary Transductive Experiments

We perform experiments on the two most commonly used benchmark knowledge graph completion datasets. One of them is WN18RR [8], which is a wordnet subset containing $40,943$ entities, $11$ relations, and $86,835$ training triples. The other is FB15k-237 [21], which is a Freebase subset containing $14,541$ entities, $237$ relations, and $272,115$ training triples. We use filtered setting for evaluation and report Mean Reciprocal Rank (MRR), Mean Rank (MR), and Hits@N $N = 10, 3, 1$. We find that the max aggregator with ConvE [8] scoring function gives the best results. Table 7 shows competitive performance on these two datasets.

| | FB15k-237 | | | | | WN18RR | | | | |
|---|---|---|---|---|---|---|---|---|---|---|
| | MRR | MR | H@10 | H@3 | H@1 | MRR | MR | H@10 | H@3 | H@1 |
| TransE [2] | .294 | 357 | .465 | - | - | .226 | 3384 | .501 | - | - |
| DistMult [27] | .241 | 254 | .419 | .263 | .155 | .43 | 5110 | .49 | .44 | .39 |
| ComplEx [22] | .247 | 339 | .428 | .275 | .158 | .44 | 5261 | .51 | .46 | .41 |
| R-GCN [17] | .248 | - | .417 | | .151 | - | - | - | - | - |
| KBGAN [3] | .278 | - | .458 | | - | .214 | - | .472 | - | - |
| ConvE [8] | .325 | 244 | .501 | .356 | .237 | .43 | 4187 | .52 | .44 | .40 |
| ConvKB | .243 | 311 | .421 | .371 | .155 | .249 | 3324 | .524 | .417 | .057 |
| SACN [18] | .35 | - | .540 | .390 | .26 | .47 | - | .54 | .48 | .43 |
| HypER [1] | .341 | 250 | .520 | .376 | .252 | .465 | 5798 | .522 | .477 | .436 |
| RotatE [19] | .338 | 177 | .533 | .375 | .241 | .476 | 3340 | .571 | .492 | .428 |
| ConvR [15] | .350 | - | .528 | .385 | .261 | .475 | - | .537 | .489 | .443 |
| VR-GCN [28] | .248 | - | .432 | .272 | .159 | - | - | - | - | - |
| RotH [4] | .314 | - | .497 | .346 | .223 | .472 | - | .553 | .490 | .428 |
| AttH [4] | .324 | - | .501 | .354 | .236 | .466 | - | .551 | .484 | .419 |
| G-MPNN (Ours) | .359 | 191 | .543 | .392 | .267 | .482 | 3412 | .546 | .498 | .446 |

Table 7: Performance of G-MPNN on binary Link prediction and several recent models on FB15k-237 and WN18RR datasets. We take the results of existing methods from their papers ('-' indicates missing). We find that G-MPNN performs comparably on FB15k-237 and WN18RR.

## 3.6 Binary Inductive Experiments

We perform experiments on the two benchmark inductive knowledge graph completion subsets released by a prior work [24]. The dataset is constructed from FB15k [2] We use filtered setting for evaluation and report Mean Reciprocal Rank (MRR), Mean Rank (MR), and Hits@N $N = 10, 3, 1$. We find that the max aggregator with TransE [2] scoring function gives the best results. Table 8 shows competitive performance on the two data subsets.

| | Subject-10 | | | | | Object-10 | | | | |
|---|---|---|---|---|---|---|---|---|---|---|
| | MRR | MR | H@10 | H@3 | H@1 | MRR | MR | H@10 | H@3 | H@1 |
| MEAN [12] | 0.310 | 293 | 0.480 | 0.348 | 0.222 | 0.251 | 353 | 0.410 | 0.280 | 0.171 |
| LSTM [13] | 0.254 | 353 | 0.429 | 0.296 | 0.162 | 0.219 | 504 | 0.373 | 0.246 | 0.143 |
| LAN [24] | 0.394 | 263 | 0.566 | 0.446 | 0.302 | 0.314 | 461 | 0.482 | 0.357 | 0.227 |
| G-MPNN (Ours) | 0.391 | 258 | 0.569 | 0.442 | 0.309 | 0.317 | 465 | 0.476 | 0.364 | 0.228 |

Table 8: Performance of G-MPNN on inductive binary Link prediction and two recent models on FB15k dataset. We take the results of existing methods from their papers. We achieve competitive results on this task.

## 3.7 Computational Complexity and Hyperparameters

let $s$ be the size of the largest hyperedge of the hypergraph. Let $d$ be the embedding dimension of hidden representation. Let $m$ be the size of the largest "neighbourhood" of a vertex (i.e. the largest number of incident hyperedges of a vertex). Then, the computational complexity of computing the hidden representation of a vertex through G-MPNN update is $O(sdm)$.

Experiments were run for 200 epochs on a GTX 1080 Ti with 12 GB RAM. The Adam optimiser was used with a learning rate of $0.01$, L2 penalty of $5e^{-4}$. Following standard practice, the size of $d$ was set to 200. The model was evaluated on the validation set and saved three epochs with the best performing checkpoint used for testing.

### 3.8  Additional Diagram for Multi-Relational Ordered Hypergraph

## Footnotes

[1] https://people.cs.umass.edu/ mccallum/data.html

[2] https://linqs.soe.ucsc.edu/data

[3] `https://aminer.org/lab-datasets/citation/DBLP-citation-Jan8.tar.bz`

[4] `https://lfs.aminer.org/lab-datasets/citation/acm.v9.zip`

[5] `https://github.com/mattbierbaum/arxiv-public-datasets/releases/tag/v0.2.0`

[6] `https://en.wikipedia.org/wiki/List_of_computer_science_conferences`