[Reviews · NeurIPS 2020]

Review 1

Summary and Contributions: This work proposed a framework that essentially generalizes MPNN and is able to process multi-relational recursive hypergraphs. This framework seems to unify many previous GNN architectures and may handle the most general graph-structured data.

Strengths: (1) The underlying formulation of this work seems to be strong and able to unify different frameworks. This is useful for readers to capture the key point of those hundreds of GNN implementation with only slight difference in their implementations. The connection between this work and previous works are explained well. (2) The work tries to capture the most complicated graph-structured data, which is challenging.

Weaknesses: (1) Many notations are missing, which makes the paper really hard to follow. For example, what are T_u, T_v in Prop. 1. What are all the bold notations in Eq. 3? The multiplication there is matrix product or vector component-wise product. (2) The whole story line is not clear. Different types of motivations come out here and there. For example, the arguments claimed in the abstract, the introduction and the overview in the Section 3 are not consistent. ------ Thank the authors for preparing the response. I have thoroughly checked the paper again and other reviewers' comments. I think the paper indeed considers some really complex and general structures in the hypergraph data. However, I still found it very hard to follow the core idea and thus hardly improve my evaluation. I will even increase my confidence score while keeping my evaluation as marginally tending to reject this paper. My specific criticisms are as the follows: 1. A lot of notations and their explanations have to be clearly highlighted, including the ones I previously mentioned and the other reviewers mentioned. 2. The motivating example is not clear enough and the formulation in G-MPNN/MPNN-R is not clear enough why they can capture the missing information in the motivating example. Also, as just the other reviewers said, in experiments, it is still not clear how to model the data that seems to be standard graphs that allows your models to indeed capture more information. 3. I still do not see why MPNN-R and G-MPNN, two very different models, should be compressed together within so limited space to make neither of them well explained. Moreover, these two models seem to have two different application domains. MPNN-R and its baselines used for standard hypergraphs. G-MPNN and its baselines used for comparison seem to be applied to heterogeneous graphs instead of natural hypergraphs. I guess that the authors in default assume that every reader knows the transformation from heterogeneous graphs to hypergraphs, which I do not think is the case. 4. I agree this paper unifies many models and generalizes many models. However, if the fundamental reason why such unification and generalization really brings benefit (not just empirical evaluation) is not shown clearly, I think such contributions are not valid and just incremental. Overall, I think that maybe there is indeed something useful in the paper. The exposition and notations are too unclear to make me (or maybe others) really appreciate this work.

Correctness: Because this work is really hard to follow (missing notations and unclear story line), I cannot justify whether the method and the claims are correct or not.

Clarity: The work is hard to follow.

Relation to Prior Work: This work is good to compare itself with previous works and positions itself. I appreciate this point.

Reproducibility: Yes

Additional Feedback:


Review 2

Summary and Contributions: this paper introduces a novel message passing neural network framework that operates over complesx, diverse relational data: (1) multi-relational ordered and (2) recursive hypergraphs, in which hyperedges can act as nodes in other hyperedges. the authors point out that this type of data in particular arises in linguistic datasets. the authors test their model on multiple benchmarks and find their model outperforms comparisons. contributions: theoretical: introduce generalized message passing function for aggregating information from multi-relational ordered hypergraphs engineering: introduce message passing function for recursively structured data based and recursive Laplacian experimental: G-MPNN and MPNN-R outperform competitors on baseline tasks

Strengths: the authors clearly demonstrate the relevance of the work with helpful examples of where recursive and multirelational ordered information informs our representations of, for example, sentences. the work is theoretically grounded, and empirical results are promising.

Weaknesses: why not run multi-relational ordered hypergraph methods [66,16,24,51,44,23] as baselines, even though they are not MPNN based? given the authors attribute the success of their model to the fact that other MPNN methods can't represent this type of information, it would be an informative baseline. how do these models compare in # parameters? architectural improvement? or simply the fact that it could access this information? in table 1, multiple entries have the same properties -- seems like there should be additional fields distinguishing them? perhaps a naive questions, but it is not clear to me from the text how these datasets were set up as multi-relational ordered hypergraphs or recursive hypergraphs. as far as i know these benchmarks are common (non-hyper) graph problems so i'm not sure where the extra information is coming from. in general, experimental details could be better explained.

Correctness: as far as i know yes.

Clarity: generally followable. however, a number of typos that make parts a bit confusing (e.g. proposition 1) edge notation on 87-88 unclear. positional mapping is also confusing. the expository figure 1 is helpful. the paper would substantially benefit from a schematic figure overviewing the G-MPNN and MPNN-R. some proofs (eg proposition 1) could be relegated to supplement. define indicence structure, incidence matrix. writing could be made more concise and tidier. further, some of the longer equations likely could be described more simply, especially where they bear on the conceptual interpretation of the paper.

Relation to Prior Work: as far as i know yes.

Reproducibility: Yes

Additional Feedback:


Review 3

Summary and Contributions: This paper introduces a generalized message passing method for multi-relational ordered hypergraph and a novel framework (MPNN-R) to manage recursively-structured datasets. It also proposes a new task of inductive learning in multi-relational ordered hypergraph. The methods are applied to semi-supervised classification and link prediction tasks and experiment on several major datasets.

Strengths: - The proposed methods are well-motivated that was proved to be effective - The experiments are well designed and detailed with good improvements over the datasets. Ablation studies are conducted on semi-supervised classification and link prediction tasks to understand the role and impact of the key parameters. - Code is provided.

Weaknesses: - I am a bit unsure of the proof for their proposition. In the major step of the proof where they represent the relation mapping (edge, direction) pair in terms of the vertex types, two important parameters T_u and T_v are not defined. Without the clear description of all notations, it is hard to be convinced that their framework is generalized. - Although the results of G-MNPP on inductive datasets are interesting, I find the authors overclaim a bit. GAATs paper [2] reported outstanding performance on transductive datasets, which outperforms G-MNPP on 4 out of 5 metrics on FB15k-237 [5] and 2 out of 5 metrics on WN18RR [6]. It would be more convincing to show the analysis of the weakness of the method. - There are many other baselines (ex: HAN [3] and MAGNN [4]), which reported outstanding performance on the semi-supervised classification tasks. It will be interesting to see the comparison of these models above with MPNN-R to clarify the effect of the principled integration of k-depth hyperedges. References: [1] Sen, P., Namata, G., Bilgic, M., Getoor, L., Galligher, B., and Eliassi-Rad, T. Collective classification in network data. AI magazine, 29(3), 2008 [2] Wang, R., Li, B.C., Hu, S.W., et al. (2020) Knowledge Graph Embedding via Graph Attenuated Attention Networks. IEEE Access, 8: 5212-5224. [3] Xiao Wang, Houye Ji, Chuan Shi, Bai Wang, Yanfang Ye, Peng Cui, and Philip S Yu. 2019. Heterogeneous Graph Attention Network. In WWW. 2022–2032. [4] Xinyu Fu, Jiani Zhang, Ziqiao Meng, and Irwin King. Magnn: Metapath aggregated graph neural network for heterogeneous graph embedding. In Proceedings of The Web Conference (WWW), page 2331–2341, 2020 [5] Kristina Toutanova, Danqi Chen, Patrick Pantel, Hoifung Poon, Pallavi Choudhury, and Michael Gamon. Representing text for joint embedding of text and knowledge bases. In Proceedings of the Conference on Empirical Methods in Natural Language Processing (EMNLP), pages 1499–1509, 2015 [6] Tim Dettmers, Pasquale Minervini, Pontus Stenetorp, and Sebastian Riedel. Convolutional 2d knowledge graph embeddings. In Proceedings of the Thirty-Second Conference on Association for the Advancement of Artificial Intelligence (AAAI), pages 1811–1818, 2018

Correctness: I have not checked the proofs thoroughly. The empirical results seem to support the theory.

Clarity: The paper is loaded with lots of notations which should be simplified. I understand it may be needed to be general and formal, but it is hard to read through.

Relation to Prior Work: Message passing on multi-relational neural networks was proposed, if not the first, in Pham, Trang, et al. "Column networks for collective classification." Thirty-First AAAI Conference on Artificial Intelligence. 2017.

Reproducibility: Yes

Additional Feedback: - To experiment results are more persuasive, the authors should elaborate the semi-supervised classification performance of MPNN-R on Citeseer dataset and Pubmed dataset [1], which are very popular on this task. - The definition of a hidden representation h_v^t is unclear in Eq. 1. Readers only know their meanings after reading Eq. 2. In line 90, the upper bound of positional mapping p (p ≤n) is not explained properly. It would be great if the authors could explain it more detail. - Typos: + Line 28 and 301: oredered -> ordered. + Line 44: aggregat -> aggregated + Line 206: defiine -> define


Review 4

Summary and Contributions: The paper presents novel extensions of message passing neural networks to very general types of graphs, namely hypergraphs with typed edges (where edges can be connected to multiple nodes and have a type) and recursive hypergraphs (where edges can be connected to other edges). The extensions follow relatively naturally from the existing literature on graph neural networks and generalize properly over existing work (with some caveats as discussed below). The experimental evaluation is quite extensive and shows notable performance gains over existing work.

Strengths: The biggest strength of the paper is its experimental evaluation, which covers seven real-world datasets and four competing methods. Another strength is that the proposed models properly generalize over a wide variety of models in the literature and extend existing generalizations logically. Finally, the topic of graph neural networks has gained increasing interest in the NeurIPS community in recent years, such that a significant portion of the audience should be interested.

Weaknesses: The main weaknesses of the paper are in its presentation and motivation. Most of the paper's notation is rather abstract and makes it difficult to understand how to implement the paper in detail. It seems to me that all the implementation details are compressed into equation (3) and the three lines below it. This is too narrow a space to fully grasp how the approach works. Further details on potential improvements in terms of presentation are covered below. With regards to motivation, the proposed generalization to typed hypergraphs and recursive hypergraphs are proper, but it is not quite clear to me from Figure 1 and the introduction alone that these graph classes are the most natural way to represent complex knowledge instead of, e.g., typed graphs with typed edges (which are, to my knowledge, the most common type of knowledge representation in triple stores etc.). In other words, the introduction does, at present, not sufficiently convince me that hyperedges add (in practice) much beyond the representational capabilities of just typed nodes and edges, which are arguably easier for users to understand. Finally, conceptually the formalization of node types via the vehicle of edge types comes with the drawback of a quadratic number of edge types (one per combination of node types), as well as a loss of interpretability. Is there a specific reason why node types are not permitted directly in this framework? /edit The authors mention in their response that node types are a trivial extension. I agree. However, it may still be beneficial to incorporate them explicitly because node types have linear memory complexity whereas edge types have quadratic complexity.

Correctness: I believe that all theoretical claims of the paper are correct and justified. Admittedly the arguments made are relatively short, but this is more a matter of presentation than correctness (see below). With regards to the experiments, I believe the methodology could be strengthened by using crossvalidation instead of repeated random splits and using statistical tests to validate the significance of performance gains. Other than that, the experimental methodology seems sound to me (and the extensive use of ablation studies is commendable).

Clarity: The paper could be, I believe, more clearly written in crucial aspects. In more detail: * Figure 1 is insufficiently clear to me. Ideally, a semantic interpretation of each hyperedge should be added. I also believe that a representation of the example in terms of a graph would add more to understanding, i.e. drawing hyperedges really as edges instead of other representations. This is also crucial to understand the appeal of hyperedges and recursive hyperedges to formalize knowledge in the first place. * In Equation (2), the notation e-v refers to the set e without the element v, I presume. This could be made clear. * In Equation 3, the operator * should be specified. I assume it refers to the element-wise product, but I'm not sure. * Most importantly, the boldface letters in Equation (3) are not properly introduced. I assume r_{e, P_e}^t is a vector representing the relation of edge (e, P_e) and p_{e, v} is a vector representing the position of node v in edge e, but this should be made clear to enable readers to actually implement the approach (or understand an existing representation). Additionally, motivation should be provided why products are used here instead of sums. * The definition of recursive hypergraphs could be made clearer. An example alternative would be to define a r-recursive hypergraph as a quadruple (V, E, P, R) (as with a multi-relational ordered hypergraph), but R is now a 'rank' function R : E \to \{0, \ldots, r\} and hyperedges e are now defined as subsets of V \cup E such that for any edge e' \in e it holds R(e') < R(e). I believe this is equivalent to the powerset formulation but more consistent with the other notation in the paper and easier to integrate with the incidence concept later on. * In the semi-supervised vertex classification, it is not clear to me why depth 0-hyperedges are modelled as vertices in the HetGNN version. If I understand correctly, each reference from a document to another is a regular edge (of depth 0), such that these could just be kept as regular edges. One just requires a different node type for authors (i.e. depth 1 hyperedges). * page 7, line 248: The description of hyperparameter search belongs to the experimental methods in the previous paragraph.

Relation to Prior Work: The paper covers prior work comprehensively and also characterizes its own relation as a generalization over much prior work correctly. I see no problems in this regard.

Reproducibility: No

Additional Feedback: With regards to reproducibility: While I applaud the inclusion of source code in the supplementary material, I believe the amount of documentation is too small and the paper is not sufficiently clear, currently, to reproduce the results. With regards to ethics: I do not believe that this work raises any direct ethical concerns in methods or data. However, the broader impact section currently only highlights the potential for positive impact, while potentially hazardous applications should be discussed as well, e.g. trying to predict criminality or credit from social networks. Such applications may reproduce and exacerbate bias and readers of the paper should be aware that the presented model should not applied naively to such tasks. typos and style: * page 1: 'Recursivesly' * page 2: 'reprsentation', 'aggregat' * page 3, line 88: $e \in E$ should probably be $e \in \mathcal{E}$ * page 3, line 89: The sentence 'The notation P is used to denote, P = ...' seems ungrammatical to me and should, perhaps, be merged with the following sentence. * page 3, line 98: The notation e_{v, w} denotes both an edge and its features, which may be confusing. It would be useful to denote the edge features perhaps with a boldface letter or some other notational variation. * page 6: defiine * In the appendix, lines 117, 119, and 121 the closing curly braces should be closing brackets

[Author Response · NeurIPS 2020]

Thanks to all the reviewers for their insightful, and constructive feedback! The reviewers raise important concerns regarding presentation, and other technical concerns.
First of all, it is encouraging to see the reviewers find that

1. The unified formulation is strong, and tries to capture the most complicated graph-structured data - **Reviewer 1 (R1)**

2. The work is theoretically grounded, and the empirical results are promising - **Reviewer 2 (R2)**

3. The methods are well-motivated, and the experiments are well-designed and detailed with ablation studies - **Reviewer 3 (R3)**

4. The experimental evaluation is the biggest strength, and the proposed models properly generalise a wide variety of models in literature - **Reviewer 4 (R4)**

**Response to Common Concern on the Proof of Proposition 1**:

**R1**, **R2**, **R3** wanted clarity on $T_u$ and $T_v$ in the proof of proposition 1. $T_u$ and $T_v$ represent vertex types of the vertices $u$ and $v$ in the heterogeneous graph $(V, E, S)$,
and they are the same as $S_u$ and $S_v$ respectively. This is certainly a typo (also acknowledged by **R3**), and we will replace $T_u$, and $T_v$ by $S_u$, and $S_v$ respectively.

**Response to Reviewer 1**: Thanks for taking the time to review the paper!

**Storyline is unclear**: As stated in the introduction, our main contributions are to unify existing MPNNs on different structures into G-MPNN, demonstrate the strong
inductive capability of G-MPNN, and extend MPNN to recursive structures. All three contributions have a common motivation e.g. Fig. 1 (acknowledged by **R2**, **R3**).

**Response to Reviewer 2**: Thanks for thoroughly reviewing the paper!

**Why not run [66,16,24,51,44,23] as baselines?**: It is not straightforward to extend those methods to the inductive setting (ability to handle unseen entities at test time).
All those methods assume that all entities are seen during training (transductive), and hence they are not baselines in the inductive experiments. Furthermore, we do use
some of them as baselines in the transductive setting (Appendix Section 2.6, Table 7) where G-MPNN performs comparably.

**How do these models compare?**: The baselines of G-MPNN have fewer parameters than G-MPNN. The additional parameters of G-MPNN are relational embeddings,
and positional embdings. The ablation study in the appendix (Section 2.1, Table 4) demonstrates the additional information are required for G-MPNN's effectiveness.

**Multiple entries have the same properties in Table 1**: Yes, in fact, multiplex networks are typically special multi-relational graphs with homophilic relations (e.g.
facebook friends, twitter followers, linkedin connections in a 3-plex network). Proposition 1 in the paper means that heterogenous graphs and multi-relational graphs are
very similar. These two points only emphasise the point that tens of recent GNNs / MPNNs have recently been proposed with small modifications, and our G-MPNN
generalises and captures the key idea in all these models (also acknowledged by **R1**).

**Unclear how datasets are setup**: Datasets for MPNN-R / recursive hypergraphs are setup from their raw sources (from their source websites such as aminer, arXiv, etc.).
For academic networks, authors are depth-1 hyperedges, and documents are vertices, and depth-0 hyperedges (please see Example 1, line 162 for details). Datasets for
G-MPNN / multi-relational ordered hypergraphs are obtained from popular multi-ary benchmarks used in existing literature [1, 2].

**Response to Reviewer 3**: Thanks for the review and the useful pointers!

**Comparison with GAAT**: Thanks for bringing attention to the GAAT paper. We will include this as a baseline in the transductive experiments (with proper credits to GAAT). One interesting weakness / extension of our paper is G-MPNN-R (Figure 1 in the paper shows a use case).

Classification error (lower is better) on recursive hypergraph datasets used in the paper

| Method | Cora | DBLP | ACM | arXiv |
|---|---|---|---|---|
| HAN | $27.24 \pm 1.9$ | $22.18 \pm 1.4$ | $23.21 \pm 2.1$ | $25.02 \pm 2.2$ |
| MAGNN | $26.78 \pm 1.5$ | $\mathbf{21.68 \pm 1.8}$ | $22.29 \pm 1.9$ | $24.23 \pm 1.8$ |
| **MPNN-R** | $\mathbf{25.34 \pm 1.5}$ | $\mathbf{21.45 \pm 1.7}$ | $\mathbf{20.32 \pm 2.1}$ | $\mathbf{22.34 \pm 1.7}$ |

**Comparison with HAN and MAGNN**: Thanks for the suggestion, we include them as baselines in the Table shown on the right.

**Citeseer, PubMed datasets**: MPNN-R is most effective for datasets containing at least depth-1 hyperedges (and higher depths). We did try to get authorship information
(depth-1 hyperedges) for Citeseer, PubMed datasets but they seem to be proprietary (not publicly accessible).

**Response to Reviewer 4**: Thanks for a very careful review, and the detailed comments!

**All the implementation details are compressed in Equation 3**: For G-MPNN, in addition to equation 3, we have given more implementation details in experimental
section 5.2, and appendix section 2.2. Still, as suggested, we will add more details (e.g. pseudocode) in the appendix to ensure that the paper is fully reproducible.

**Motivation for typed hyperedges over typed edges**: While we agree that typed edges are popular / common knowledge representations in triple stores, typed hyperedges,
on the other hand, have their own benefits over typed edges (e.g. more flexible organisation of multi-ary relational facts, more representative than binary relations, etc.)
and have been a recent research topic of interest esp. in practice [1,2]. In a separate work [3], it has been shown that a few sentence types such as claims about claims in
natural language (e.g. A claimed that B claimed C) can flexibly be represented by recursive hypergraphs with typed hyperedges rather than graphs with typed edges. As
suggested, we will expand the introduction section with more motivation.

**Is there a specific reason why node types are not permitted directly in this framework?**: Given the G-MPNN formulation, and a large body of work on MPNNs
with node types, it is quite straightforward to permit node types (without the vehicle of edge types) in a generalised MPNN (e.g. use a separate function for node types).

**Statistical tests to validate the significance of gains**: Thanks for the suggestion (we will add statistical tests). **R3** has suggested two more recent baselines for MPNN-R
(please see the table above). Based on a Welch t-test, the p-value on DBLP is $0.35$, and a small p-value on the other three (less than $0.0005$) validates the significance.

**Figure 1 is insufficiently clear**: Thanks for the important comment on improving the understanding of the Figure. Due to space limitations, we have attempted to express a subgroup of words (*to actors Timothy and Wanda*) through typed edges (shown on the right). As suggested, we will include the representation of the example in terms of a graph (with typed edges).

**Explain boldface letters in Equation 3, $*$, and $e - v$**: The detailed comments are highly appreciated! We will fix them. Yes, $*$ represents element-wise product.

**Motivation should be provided for product over sums**: This follows directly from knowledge representation literature where multiplicative interactions are preferred to additive interactions (e.g. [4]). Our
specific multiplicative choice of G-MPNN also generalises existing formulations shown in lines 147-149.

**Definition of recursive hypergraphs could be made clearer**: Thanks for the alternative definition. We agree that it has important benefits. A small caveat is that $R$ is
already used for MPNN readout, and $\mathcal{R}$ is used for relations (we will see how best to use the alternative with intuitive notations without overloading existing notations).

**Depth-$0$ hyperedges as vertices in HetGNN**: We have explaned a general method of modelling a recursive hypergraph with a heterogenous graph. For the specific
example of academic networks used in the paper, we agree that both vertices and depth-0 hyperedges represent the same entities with the same type (documents).

Thanks to all the reviewers again for the comments! All typos, concerns on ethics, and other minor concerns will be fixed.

[1] Bahare Fatemi, Perouz Taslakian, David Vazquez, and David Poole. Knowledge hypergraphs: Prediction beyond binary relations. In IJCAI'20.

[2] Jianfeng Wen, Jianxin Li, Yongyi Mao, Shini Chen, and Richong Zhang. On representation, embedding of knowledge beyond binary relations. In IJCAI'16.

[3] Telmo Menezes, Camille Roth. Semantic Hypergraphs, 2019

[4] Embedding Entities and Relations for Learning and Inference in Knowledge Bases, In ICLR'15


[Meta-Review · NeurIPS 2020]

The paper proposes to generalize message passing neural networks (MPNN) for modeling multi-relational, recursive and ordered hypergraphs, which is theoretically grounded and challenging topic. The authors tested the proposed framework on semi-supervised node classification and link prediction tasks on several (7) benchmark datasets, where the proposed methods outperformed other baselines consistently. Although some reviewers point out the presentation issues with unclear notations and suggest ways to improve and more baselines in addition, the merits of the paper outweigh the drawbacks and acceptance is recommended. We strongly encourage the authors to take the reviewers’ feedback into account in their revision of the paper.